# Predictors of Couple Burnout among Turkish Married Individuals

**DOI:** 10.3390/bs14070561

**Published:** 2024-07-03

**Authors:** Nursel Topkaya, Ertuğrul Şahin, Cansu Terzioğulları Yılmaz, Nuray Aşantuğrul

**Affiliations:** 1Department of Guidance and Psychological Counseling, Faculty of Education, Çanakkale Onsekiz Mart University, Çanakkale 17000, Türkiye; 2Department of Guidance and Psychological Counseling, Faculty of Education, Amasya University, Amasya 05100, Türkiye; ertugrulsahin@amasya.edu.tr; 3PsikoSamsun Counseling and Education Center, Samsun 55070, Türkiye; cansu_terziogullari@hotmail.com; 4Independent Researcher, Amasya 05100, Türkiye; nry.firat@hotmail.com

**Keywords:** couple burnout, relationship self-efficacy, self-compassion, happiness, sociodemographic factors, Türkiye

## Abstract

Couple burnout has been linked to several negative consequences for both individuals and couples. Identifying the factors that predict couple burnout is essential for developing effective interventions to prevent or lessen its detrimental impact on marital relationships. The aim of this cross-sectional study was to investigate sociodemographic factors, relationship self-efficacy, happiness, and self-compassion as predictors of couple burnout in Turkish married individuals. A convenient sample of 401 married individuals completed a questionnaire that comprised a Personal Information Form, Couple Burnout Measure—Short Version, Relationship Self-Efficacy Scale, Self-Compassion Scale, and Single-Item Happiness Scale. The data were analyzed using descriptive statistics, Pearson’s product-moment correlation analysis, linear multiple regression analysis, and relative importance analyses. The results of this study suggest that being women, having a higher number of offspring, and lower levels of relationship self-efficacy, self-compassion, and happiness were significant positive predictors of couple burnout among married individuals. The type of marriage, monthly income, and duration of marriage were not significant predictors of couple burnout. Moreover, the results of the relative importance analyses consistently demonstrated that happiness was the strongest predictor of couple burnout among married individuals. The research findings demonstrate the multidimensional nature of couple burnout and provide a more nuanced understanding of its predictive factors. These results have potential implications for the development of evidence-based and targeted interventions in relationship education programs.

## 1. Introduction

Although general burnout has been the subject of much systematic investigation in recent decades, a growing body of the literature highlights the significance of exploring various forms of burnout within specific life domains. One type of burnout that has received increasing attention in intimate relationships is couple burnout (also used interchangeably in the literature with relationship burnout or marital burnout) due to a number of factors, including the increasing number of dual-earner couples, the increasing demands of work and family life, and the changing nature of intimate relationships [1,2,3,4]. Couple burnout refers to a state of physical, emotional, and cognitive exhaustion that arises from the discrepancy between the expectations and realities of a relationship [3]. Research indicates that couple burnout is a complex response to common challenges and chronic stressors faced by couples in romantic relationships, such as financial strain, work–life balance issues, and childcare responsibilities [2,3,4,5]. Couple burnout is also a cumulative process and, as it progresses, can manifest itself with complex physical, cognitive, and emotional symptoms such as chronic fatigue, negative feelings about the relationship and its future, hopelessness, disappointment, feelings of detachment, resentment, and decreased intimacy [3,6,7].

Although research on couple burnout is still in its early stages, existing studies in clinical and non-clinical samples suggest that higher levels of couple burnout are associated with several outcomes that can negatively affect individuals’ quality of life, mental health, and marital life, such as lower marital satisfaction, lower levels of trust between couples, perceptions of lower spousal support, difficulty meeting basic psychological needs, lower levels of resilience, higher levels of perceived stress and anxiety, higher levels of conflict and criticism in family communication, and higher divorce proneness [2,5,8,9,10,11,12,13]. In addition to harmful effects on the mental health of couples, research also suggests that couple burnout negatively affects the healthy development of children [14]. According to Pines [3], couple burnout can eventually lead to psychological problems, emotional divorce, and legal divorce by reducing feelings of intimacy, love, and compassion between couples. Considering the association of couple burnout with negative outcomes in marital life and mental health, identifying the associated factors can provide useful information for mental health professionals to identify individuals at risk of couple burnout, develop protective and preventive mental health services for individuals with couple burnout, and develop effective marriage and family counseling interventions.

Although there are different types of intimate relationships, such as engaged, dating, or cohabiting couples, marriage is one of the most common types of intimate relationships and requires the fulfillment of a set of family development tasks that may increase the level of couple burnout, such as creating a satisfactory marital environment, adapting to the pregnancy process, having children, meeting the needs of children during the education period, establishing relationships with each other’s relatives, and adapting to retirement [15]. In addition to these family development tasks, marriage is a complex relationship that involves high levels of intimacy, commitment, and interdependence among couples [16]. This can make couples more vulnerable to stress and burnout, especially if they are unable to meet each other’s needs or effectively manage their expectations [3]. Therefore, marriage is an important type of relationship in which factors associated with couple burnout need to be investigated.

Theoretical explanations and previous studies on couple burnout suggest that couple burnout may result from a complex combination of several demographic, familial, social, and cultural factors [4,5,7,9,10,13]. One factor that may help protect couples from couple burnout is relationship self-efficacy, defined as the belief in one’s ability to form and sustain romantic relationships [17]. Individuals with high relationship self-efficacy are more likely to engage more actively in their relationships, exhibit greater resilience in the face of challenges, exert increased effort to overcome obstacles, and sustain their relationships [17,18]. Therefore, relationship self-efficacy can be considered an important variable that can reduce couple burnout among married individuals. Although there is a growing body of research on the relationship between general self-efficacy and burnout, there is a paucity of research on the relationship between relationship self-efficacy and couple burnout [12]. Furthermore, because the study conducted by Moravejjifar et al. [12] only examined pregnant mothers, the generalizability of their findings to married men and non-pregnant married women is low. Therefore, more research is needed to examine the relationship between relationship self-efficacy and couple burnout among married individuals.

Another factor that may help couples to successfully maintain their marriage and protect them from couple burnout may be self-compassion. Self-compassion refers to perceiving one’s personal experiences as common human experiences, viewing difficulties, failures, pain, and inadequacies as part of being human, and accepting that one and all others are worthy of being treated with compassion [19]. Individuals may experience negative emotions such as pain, anger, sadness, self-pity, and burnout throughout the family life cycle. Coping with such situations and persevering through problems can be challenging for couples. Individuals with higher levels of self-compassion may have more compassionate attitudes toward themselves when faced with such negative situations [19]. To the best of our knowledge, no study has yet examined the relationship between self-compassion and couple burnout among married individuals; however, studies focusing on the benefits of self-compassion in romantic relationships indicate that self-compassion may be negatively related to couple burnout. Neff and Beretvas [20] found that highly self-compassionate individuals in romantic relationships also had high relationship subjective well-being, which consisted of a sense of self-worth, positive affectivity, the ability to express one’s emotions, and authenticity. In addition, Jacobson et al. [21] found that highly self-compassionate individuals in romantic relationships had higher levels of couple adjustment and relationship satisfaction. In a comprehensive literature review, Latren et al. [22] also concluded that self-compassion may help individuals build stronger and more supportive relationships by increasing their willingness to communicate openly and honestly about their needs and feelings in close interpersonal relationships. Self-compassion may also help couples cope with stress more effectively by helping them maintain a positive outlook on their relationship, even during difficult times [20]. Based on these findings, higher levels of self-compassion may be a relational factor that helps prevent and reduce couple burnout in married individuals.

Couple burnout is a multifaceted phenomenon influenced by various factors. While individual traits like self-efficacy and self-compassion have been explored in relation to couple burnout, it is equally important to examine the role of sociodemographic variables. These factors, encompassing gender, education level, duration of marriage, type of marriage, number of children, and monthly income, may contribute to or mitigate the risk of burnout among married individuals. Additionally, subjective well-being measures like happiness could potentially be associated with the experience of couple burnout. Moreover, previous research has yielded mixed findings regarding the influence of these sociodemographic factors, highlighting the need for further investigation to better understand their correlations with couple burnout [2,4,5,7,9,10,11]. Previous studies have revealed that couple burnout differs between women and men; women are more likely to experience couple burnout than men [4,6,9,13]. On the other hand, studies examining the relationship between other sociodemographic factors such as education level, duration of marriage, type of marriage, and number of children and couple burnout have reported inconsistent findings [2,9,10,12,13]. Although some studies have found no significant relationship between education level and couple burnout in married individuals [2,7,9], other studies have reported that education level is negatively associated with couple burnout [13]. Nejatian et al. [2] found that women who entered into a forced marriage were more likely to experience couple burnout than women who did not enter into a forced marriage. Pamuk and Durmuş [13] also revealed that couple burnout was higher in parent-arranged marriages than in acquainted marriages. While some studies found no significant relationship between the number of children and couple burnout in married individuals [2,9,11], other studies revealed that the number of children was positively associated with couple burnout [10,13]. Although Nejatian et al. [2] reported that the duration of marriage was positively correlated with couple burnout, other studies found no significant relationship between the duration of marriage and couple burnout [9,11].

Another demographic variable examined in this study is monthly income. Although some studies found no significant relationship between perceived income and couple burnout in married individuals [7], other studies reported that individuals with high income were less likely to experience couple burnout than individuals with low income [5]. Lastly, we investigate the potential relationship between happiness and couple burnout, an association that has not been directly explored in previous research to the best of our knowledge. Happiness and couple burnout may share a bidirectional relationship, where lower levels of happiness could contribute to higher couple burnout, and conversely, couple burnout could diminish an individual’s overall sense of happiness and well-being. Happiness is a subjective state of well-being that encompasses positive emotions, life satisfaction, and a sense of meaning and purpose [23,24]. It is a crucial aspect of mental health and overall quality of life [23,24,25,26]. When individuals experience high levels of happiness, they may be better equipped to handle the challenges and stressors that can arise in a marital relationship, potentially reducing the risk of couple burnout. Conversely, a lack of happiness and a predominance of negative emotions could exacerbate the emotional exhaustion and strain associated with couple burnout. According to Pines [3], couple burnout can lead to psychological problems and relationship dissatisfaction. It may also increase negative emotions such as anger, sadness, and resentment over time [3,6,7]. Therefore, it is plausible that individuals with lower levels of happiness may be more susceptible to experiencing higher levels of couple burnout, highlighting the importance of examining this potential association.

Because most previous studies on correlates of couple burnout were conducted in Iran and Western countries [2,7,8,11,12], it is crucial to evaluate the validity and generalizability of prior research findings in different cultural contexts, including Turkish society. Turkish culture encompasses unique norms, values, and relationship dynamics that may impact the experience of couple burnout among married individuals. Specifically, Turkish culture places great importance on marriage and family [27]. This can put pressure on couples to remain together even when facing difficulties, increasing their risk of experiencing couple burnout. Additionally, traditional gender roles persist in Turkish society, with men expected to be the breadwinners and women responsible for childcare, household chores, and caring for elderly relatives [28,29]. The rigid gender roles prevalent in Turkish culture can result in an uneven distribution of responsibilities, relationship imbalances, and stress. Additionally, there is a significant emphasis on family ties and extended family relationships within Turkish culture, as highlighted by Ataca [28]. Children are expected to maintain close ties with their parents and extended family throughout their lives. This can result in increased responsibilities, obligations, and conflicts, which may contribute to higher levels of couple burnout among married individuals in Türkiye. Additionally, Türkiye has a relatively high unemployment rate and those employed often work long hours and experience elevated levels of stress [30]. Furthermore, Türkiye has encountered economic hardships (e.g., high inflation) in recent years, leading to financial stress and anxiety among couples. These economic challenges may contribute to overwhelming feelings and stress among married Turkish couples, potentially leading to an increased risk of couple burnout. Thus, the aim of this study is to investigate the predictive power of sociodemographic factors, relationship self-efficacy, happiness, and self-compassion on couple burnout among Turkish married individuals.

## 2. Methods

### 2.1. Research Design

The present study employed a cross-sectional research design to investigate the predictors of couple burnout among married individuals in Türkiye. A cross-sectional design involves collecting data on the variables of interest from the study participants at a single time point, providing information about temporal relationships between the variables.

### 2.2. Settings

This study was conducted in Samsun, Türkiye. Samsun is a metropolitan city in the central Black Sea region of Türkiye, with a population of 1,107,991 as of the 2022 census results [31]. As a metropolitan city, Samsun has a diverse population in terms of socioeconomic status, education levels, and occupations, which allows for a varied sample of married individuals. Data collection took place over a period of approximately three months, from September 2022 to December 2022. The questionnaire packet consisted of a brief cover letter explaining the aim of the study, a written informed consent form, and a copy of the measures. Participants were recruited using a combination of online and in-person methods. Online announcements were posted on popular social media platforms and forums (e.g., Instagram, X, and Facebook) targeting married individuals. We recruited individuals for face-to-face data collection from local community centers, universities, other public areas, and word-of-mouth referrals. Married individuals who were willing to participate in the study online were invited to complete the questionnaire using Google Forms.

### 2.3. Sampling Method and Sample Size

The participants were recruited using the convenience sampling method. This method was chosen due to its practicality and accessibility in reaching married individuals in Samsun. To mitigate potential biases associated with convenience sampling, efforts were made to diversify recruitment sources, including community centers, social media platforms, and local organizations. The required sample size was determined using a priori power analysis conducted with the G*Power 3.1 software [32].

### 2.4. Power Analysis

A priori power analysis was conducted to determine the minimum sample size necessary for this study. Power analysis was conducted using a medium effect size (*R*^2^ = 0.13) and a minimum power of 0.99, with a significance level of 0.01, taking into account 20 potential predictor variables in a linear multiple regression model. The results of the power analysis indicated that a minimum sample size of 346 participants was required to achieve the desired level of power [32]. The medium effect size was selected based on the results of previous studies examining the association between sociodemographic characteristics, relationship self-efficacy, and couple burnout [2,4,6,9,10,12]. To better represent the sample and account for possible missing values and incomplete questionnaires, we aimed to recruit a minimum of 380 participants.

### 2.5. Participants

The participants of this study consisted of 401 married individuals. Participants eligible for inclusion in the study were required to be currently married, at least 17 years old, fluent in Turkish, and residing in Samsun. According to Article 124 of the Turkish Civil Code (No. 4721), individuals who have reached the age of 17 years may be granted permission to marry with the consent of their legal guardians.

### 2.6. Variables

Couple burnout served as the dependent variable, while the independent variables included sociodemographic characteristics, relationship self-efficacy, self-compassion, and happiness in this study. Consistent with the study’s purpose, this study did not take into account potential confounders or effect modifiers (e.g., moderating and mediating variables) in the analysis and examined relationships between the specified predictor variables and couple burnout.

### 2.7. Bias

Consistent with the recommendation of Podsakoff et al. [33], several measures were implemented to address potential sources of bias in this study. To mitigate selection bias inherent in convenience sampling, we diversified our recruitment sources, including community centers, universities, public areas, and online platforms, aiming to increase the sample’s representativeness. To reduce response bias, particularly social desirability bias, participants were assured of anonymity and confidentiality, and questionnaires were self-administered. Additionally, we used scales with reverse-scored items to further minimize response bias and encourage thoughtful, accurate responses from the participants. We utilized validated Turkish versions of all the scales to ensure cultural appropriateness and measurement accuracy, though we acknowledge the potential limitations of the single-item happiness scale.

To minimize researcher bias, the data analysis was conducted using standardized statistical procedures. We included both male and female participants to ensure gender representation, although the sample had a higher proportion of women. While we incorporated several sociodemographic and psychological variables in our analysis, we recognize that other unmeasured factors could influence couple burnout, a limitation acknowledged in our discussion. These efforts collectively aimed to enhance the validity and reliability of our findings, although we acknowledge that completely eliminating bias in cross-sectional research is challenging.

### 2.8. Measures

*Sociodemographic characteristics.* A Personal Information Form was used to determine the participants’ sociodemographic characteristics. Participants were asked to provide information about their gender, education level, duration of marriage, type of marriage, number of children, and monthly income in this form.

*Couple Burnout.* The Couple Burnout Measure—Short Version (CBM-SF; Pines et al. [4]) was used to measure couple burnout among the married individuals. The CBMS is a self-report measure that was developed to assess the level of couple burnout in individuals in all types of relationships, including married, engaged, dating, or couples [4]. The CBM-SF is a 10-item scale that measures the physical, emotional, and cognitive symptoms of couple burnout. Each item is rated on a 7-point Likert scale ranging from 1 (*never*) to 7 (*always*). The Turkish adaptation, validity, and reliability study of the CBM-SF was conducted by Çapri [6]. The Turkish adaptation of the CBM-SF had a single-factor structure, similar to the original scale [6]. Additionally, its test–retest reliability and internal consistency reliability were reported as 0.88 and 0.91, respectively [6]. The scale score is calculated by averaging item responses, with a possible range of 1 to 7. Higher scores indicate greater couple burnout. The coefficient alpha reliability of the CBM-SF was calculated as 0.93 in this study, which indicates a very high degree of internal consistency among the items on the scale [34]. A sample item from the CBM-SF is “Helpless”.

*Relationship Self-Efficacy.* The Relationship Self-Efficacy Scale (RSES; Lopez et al. [18]) was used to measure the relationship self-efficacy levels of the married individuals. Akan [35] adapted the scale into the Turkish language and conducted its validity and reliability studies. The original version of the scale consists of 25 items, whereas the Turkish version consists of 16 items. The three-factor structure of the original scale was also maintained in the Turkish version [35]. Each item is rated on a 5-point scale ranging from 1 (*not at all sure*) to 5 (*completely sure*) in the Turkish version of the scale. No items on the scale are reverse-coded. The total scale score was used in the current study to measure the overall relationship self-efficacy levels of the married individuals. Possible scores range from 16 to 80. Higher scores indicate greater relationship self-efficacy. The alpha internal consistency coefficient of the Turkish version of the scale was reported as 0.81 by Akan [35]. The coefficient alpha reliability of the RSES was calculated as 0.84 in the current study, which indicates a high degree of internal consistency among the items on the scale [34]. A sample item of the scale is “Accept your partner’s affection freely and comfortably”.

*Self-Compassion.* The Self-Compassion Scale (SCS; Neff [36]) was used to measure the self-compassion levels of the married individuals. The original version of the scale has 26 items that measure six factors: self-kindness, self-judgment, common humanity, isolation, mindfulness, and over-identification. The adaptation of the SCS into the Turkish language and its validity and reliability studies were conducted by Deniz et al. [37]. Although the original SCS has 26 items that measure six factors, the Turkish version of the SCS has 24 items that measure a single factor structure. Each item is rated on a 5-point Likert scale ranging from 1 (*almost never*) to 5 (*almost always*). The Turkish version of the SCS includes 11 reverse-scored items. Total scores can range from 24 to 120, with higher scores indicating greater self-compassion. The internal consistency coefficient and the test–retest reliability of the Turkish version of the SCS were reported as 0.89 and 0.83, respectively [37]. Cronbach’s alpha internal consistency coefficient of the SCS was calculated as 0.85 in this study. A sample item from the SCS is “When things are going badly for me, I see the difficulties as part of life that everyone goes through”.

*Happiness.* The Single-Item Happiness Scale (SIHS; Topkaya et al. [38]) was used to measure the happiness levels of the participants. Single-item happiness scales are frequently used in psychology studies to measure the happiness levels of participants, as in this study [25,26]. In addition to substantial evidence for face and content validity, previous studies examining the psychometric properties of SIHS scores also revealed strong evidence for its convergent and divergent validity with meaningful correlations in the hypothesized directions with depression, anxiety, stress, life satisfaction, self-esteem, and self-efficacy scores [25,26]. Specifically, the participants were asked to answer the following question: “Taken your life as a whole, how would you rate your happiness?” The SIHS is a 10-point Likert-type scale that asks participants to rate their happiness on a scale from 1 (*very unhappy*) to 10 (*very happy*). Possible scores on this scale can range from 1 to 10, with higher scores indicating greater level of happiness.

### 2.9. Procedure

Ethical approval was obtained from the Ondokuz Mayıs University Social Science and Humanities Institutional Review Board prior to commencing this study. We conducted a pilot study with seven married individuals to test the applicability and clarity of the measures and to identify any ambiguous items. All participants reported that all questionnaire items were clear and easy to read and understand. In accordance with ethical principles, participants were informed about the confidentiality of their data, their right to withdraw from the study at any time without any sanction, the purpose and importance of the research, and the voluntary nature of the study. All married individuals provided informed consent and voluntarily participated in the study without any form of incentive or reward. Face-to-face administered scales took approximately 15 min for the participants to complete.

### 2.10. Statistical Analysis

All statistical analyses were performed using the IBM SPSS Statistics 26 for Windows. We performed preliminary analyses to assess the accuracy of our data, identify any missing values, detect univariate and multivariate outliers, and verify the assumptions required for the correlation and regression analysis [34,39]. Examination of the minimum and maximum values and frequencies of the categorical and continuous variables indicated that all data values were within the expected range, indicating data accuracy. However, we recategorized the education level and monthly income variables to facilitate statistical analyses after examining the frequency distributions. Because all sociodemographic questions and scale items were designated as mandatory in Google Forms and the questionnaires were carefully checked by the researchers during in-person administration, there were no missing data in the dataset. Although data were initially collected from 410 married individuals, 5 univariate outliers and 4 multivariate outliers were identified during the outlier examination and subsequently excluded from the dataset [39]. Therefore, statistical analyses were performed using the remaining 401 usable data points.

Descriptive statistics, such as the mean, standard deviation, frequencies, and percentages, were used to provide information about the sociodemographic characteristics of the married individuals. Pearson’s product-moment (point-biserial) correlation analyses were conducted to determine the strength and direction of the linear relationship of couple burnout with gender, marital type, education level, marital duration, number of children, relationship self-efficacy, self-compassion, and happiness levels. Multiple linear regression analysis was used to identify independent variables that predict couple burnout. Since education level and marital duration were measured categorically with more than two levels in this study, we recoded them as dummy variables before conducting correlation or regression analyses. The first category of either education level or marital duration variable was used as the reference category in the multiple linear regression analysis. We also conducted preliminary analyses to test the assumptions of normality, linearity, homoscedasticity, and multicollinearity in the relevant analyses and found that these assumptions were met [34,39].

To determine the relative importance of each predictor variable in the multiple linear regression analysis, we employed three useful measures, including Pratt’s standardized relative importance value (*d_p_*; Wu et al. [40]), the standardized regression coefficient (β), and the semi-partial correlation coefficient (*sr*). Pratt’s *d_p_* provides information about the contribution of each independent variable to the explained variance ratio, which is the proportion of the variance in the dependent variable that is explained by the independent variables in the regression equation [40]. The standardized regression coefficient provides information about the standard deviation change that will occur in the dependent variable when one standard deviation change occurs in the independent variable [39]. Finally, the semi-partial correlation coefficient provides information about the unique contribution of each independent variable to the regression equation. The square of the semi-partial correlation coefficient indicates the expected change in the proportion of the explained variance when the relevant variable is included in the regression equation or when controlling for the effect of other variables [34,39].

The effect size classification proposed by Cohen [41] was employed to interpret correlation coefficients and the proportion of the explained variance (*R*^2^). These effect size guidelines have been widely adopted in the behavioral and social sciences as a means of evaluating the practical significance of observed effects. Absolute values between 0.00 and 0.29 were considered small, absolute values between 0.30 and 0.49 were considered moderate, and absolute values of 0.50 or greater were considered large effect sizes for correlation coefficients as per this classification. With respect to the proportion of the explained variance (*R*^2^), values below 0.12 were considered small, values between 0.13 and 0.25 were considered moderate, and values of 0.26 or greater were considered large effect sizes. The data that support the findings of this study are available from the Open Science Framework (osf.io/sb7xp). A significance level of *p* < 0.01 was used in all statistical analyses to effectively control the Type I error rate.

## 3. Results

### 3.1. Participants

Among the married individuals, 261 (65.1%) were female and 140 (34.9%) were male. The age of the participants ranged from 17 to 65 years, with a mean age of 37.51 years (*SD* = 8.26). The majority of the married individuals had an acquainted marriage (*n* = 343; 85.5%) and reported a monthly income of USD 528 or more as of the date of the survey (TRY 7001 or more; *n* = 273; 68.1%). In terms of education level, 19% of the participants (*n* = 76) had high-school or below education, 12% of the participants (*n* = 48) had an associate degree, 54% of the participants (*n* = 216) held an undergraduate degree, and 15% of the participants (*n* = 60) had a graduate degree. With regard to duration of marriage, the largest proportion of the participants (29.2%; *n* = 117) had been married for 6–10 years, followed by those married for 1–5 years (22.4%; *n* = 90), 11–15 years (20.0%; *n* = 80), 21 years or more (15.9%; *n* = 64), and 16–20 years (12.5%; *n* = 50). Finally, the number of children varied, with 15% of the participants (*n* = 60) having no children, 30.9% (*n* = 124) having one child, 44.4% (*n* = 178) having two children, and 9.7% (*n* = 39) having three children.

### 3.2. Pearson’s Product-Moment Correlation Analyses

The results of the Pearson product-moment correlation analyses conducted to examine the relationship between couple burnout and gender, type of marriage, monthly income, education level, duration of marriage, number of children, relationship self-efficacy, self-compassion, and happiness in married individuals are shown in Table 1.

As seen in Table 1, the couple burnout scores were weakly and positively correlated with being a woman (*r* = 0.22), having a marriage duration of 11–15 years (*r* = 0.13), having a marriage duration of 16–20 years (*r* = 0.16), and the number of children (*r* = 0.22) among the married individuals. Additionally, a low negative correlation was found between the couple burnout scores and monthly income (*r* = −0.18). Furthermore, the relationship self-efficacy (*r* = −0.42) and self-compassion (*r* = −0.38) scores were moderately negatively correlated with the couple burnout scores. A high negative correlation was also found between the couple burnout scores and happiness scores (*r* = −0.66). On the other hand, the couple burnout scores were not correlated with the type of marriage (*r* = 0.08), having a high-school or below education level (*r* = 0.09), having an associate degree (*r* = −0.00), having a graduate degree (*r* = −0.02), having an undergraduate degree (*r* = −0.02), having a marriage duration of 1–5 years (*r* = −0.10), having a marriage duration of 6–10 years (*r* = −0.07), or having a marriage duration of 21 years or more (*r* = −0.09).

### 3.3. Results of Linear Regression Analysis and Relative Importance Analyses

Standard multiple linear regression analysis was performed to determine the best linear combination of sociodemographic factors, relationship self-efficacy, self-compassion, and happiness for predicting couple burnout among the married individuals. The change statistics for the multiple linear regression model are shown in Table 2, and the multiple linear regression results are shown in Table 3.

As shown in Table 2, the multiple linear regression model that was established to predict the couple burnout scores of the married individuals was statistically significant (*F*(14, 386) = 33.62, *p* < 0.001, Δ*R*^2^ = 0.55). The multiple linear regression model had a high level of effect size and explained about 55% of the variance in the couple burnout scores among the married individuals.

As seen in Table 3, being a woman (β = 0.16, *t*(386) = 4.44, *p* < 0.001) and the number of children (β = 0.14, *t*(386) = 3.14, *p* < 0.01) were found to be significant positive predictors of couple burnout among the married individuals. In contrast, the relationship self-efficacy (β = −0.16, *t*(386) = −4.01, *p* < 0.001), self-compassion (β = −0.13, *t*(386) = −3.32, *p* < 0.001), and happiness (β = −0.52, *t*(386) = −12.97, *p* < 0.001) scores were found to be significant negative predictors of the couple burnout scores. The type of marriage, monthly income, and duration of marriage were not significant predictors of couple burnout in the married individuals. The results of the relative importance analyses consistently demonstrated that happiness was the strongest predictor of couple burnout, independently accounting for approximately 62% of the changes in the *R*^2^ value. Overall, the results of the multiple linear regression analysis suggest that married individuals who are female, have a higher number of children, and have lower levels of relationship self-efficacy, self-compassion, and happiness are more likely to have higher couple burnout.

## 4. Discussion

This study investigated the association between couple burnout and sociodemographic factors and the levels of relationship self-efficacy, self-compassion, and happiness among married individuals. The findings indicated a significant gender difference, with women exhibiting significantly higher levels of couple burnout than men. This finding supports the results of previous studies indicating a positive association between being a woman and couple burnout [4,6,9,13]. For example, Pines et al. [4] found that women have higher levels of couple burnout than men among Israeli working, sandwiched-generation couples. Similarly, Çapri [6] found that women were more likely to experience couple burnout than men among married working couples. Cultural differences in gender roles influence the perspectives of both women and men regarding marriage as well as their expectations of marriage and the roles and responsibilities associated with it. In Turkish culture, married women are generally expected to undertake household chores such as cooking, laundry, sewing, daily/weekly house cleaning, childcare, and dishwashing [29]. In addition, in Turkish culture, married women are expected to fulfill the demands of being a wife, including providing emotional support and companionship to their husbands, helping husbands manage family finances, and respecting and taking care of elderly family members. They are also expected to cope with job-related stress if employed. Thus, Turkish married women may experience higher levels of couple burnout than Turkish married men because of the significant responsibilities placed on them in Turkish family life.

The number of children was found to be positively associated with couple burnout in the married individuals in this study. This finding was consistent with the results of previous studies that demonstrated a positive association between the number of children and couple burnout in married individuals [2,10,13,42]. For example, Çapri and Gökçakan [42] found that the number of children was a significant positive predictor of couple burnout among married men. Similarly, Nejatian et al. [2] also found that the number of children was positively correlated with physical symptoms of couple burnout among married women. Married couples become parents after having children. Parenting is a lifelong and demanding responsibility that involves fulfilling various duties and responsibilities. As parents, individuals not only take care of their children’s day-to-day needs but also strive to create the necessary conditions and provide resources to ensure their children’s healthy development [15]. Therefore, caring for and raising children can be a stressful experience for couples. On the other hand, while transitioning to parenthood can be a rewarding experience and bring feelings of love, joy, and fulfillment for couples, research suggests that it can also be a major life challenge with a substantial impact on various aspects of a person’s life, such as relationships, work, finances, and physical and mental health [15,43]. Married individuals may encounter challenges in balancing their responsibilities and fulfilling their roles as spouses while simultaneously addressing the demands of childcare and meeting the needs of their child following the transition into parenthood [43]. As the number of children increases, parents can often find themselves investing more effort in supporting their children’s developmental processes and can experience increased stress related to childcare. Increased caregiving responsibilities can lead to difficulty in meeting the demands of the spouse and parent roles, which may be associated with higher levels of couple burnout.

The findings revealed that relationship self-efficacy is negatively associated with couple burnout, such that married individuals with higher confidence in their ability to manage and maintain a satisfying relationship tend to have lower couple burnout. The results of this study are consistent with those of a previous study among pregnant women indicating that relationship self-efficacy is negatively related to couple burnout [12]. This study also extends these findings to the broader population of married individuals and suggests that relationship self-efficacy may be an important protective factor against couple burnout regardless of gender. According to relationship self-efficacy theory, individuals with high relationship self-efficacy are better able to handle conflicts successfully, reach a consensus on their differences with their partners, and express their expectations regarding their rights within the relationship [17,18]. Furthermore, these individuals are more likely to exert more effort in overcoming the challenges they encounter within their relationships to maintain a satisfying relationship [18]. High relationship self-efficacy can therefore prevent couple burnout by helping married individuals to engage in relationship-enhancing and strengthening behaviors which, in turn, can lead to having a happier and more satisfying relationship.

Self-compassion was found to be negatively associated with couple burnout among the married individuals in this study. These findings are consistent with studies that have examined the relationship between self-compassion and general burnout levels. For example, Abdollahi et al. [44] found that low self-compassion was a significant predictor of higher levels of burnout among health care professionals. Previous studies have shown that individuals with high self-compassion tend to experience fewer psychological problems, including depression, anxiety, and stress; experience higher levels of positive emotions; demonstrate effective coping and emotion regulation skills; and have better overall physical, psychological, and cognitive health (for a review, see Neff [19]). Studies examining the benefits of high self-compassion in close interpersonal relationships have also found that individuals with high self-compassion have a better understanding of their romantic partners, use constructive conflict resolution strategies, and engage in relationship-protective behaviors [20,22]. Accordingly, high self-compassion may protect individuals from couple burnout by helping them engage in healthier and more positive behaviors toward their partners and to be more attentive to the needs of their relationship.

The findings of the current study revealed a negative association between happiness and couple burnout such that happy married individuals tended to experience lower couple burnout. Moreover, the relative importance analyses revealed that happiness was the strongest predictor of couple burnout, underscoring its crucial role in preventing relationship burnout. These findings are consistent with theoretical expectations [3] as well as findings indicating a negative association between happiness and burnout levels [45,46]. According to Pines [3], the primary objective of individuals in a marriage is to achieve personal and mutual happiness. Pines also posits that when individuals’ expectations about their spouse and marriage are fulfilled, they tend to experience greater happiness and lower couple burnout. On the other hand, previous research has shown that happy individuals are generally healthier, have stronger and more satisfying relationships, and are more successful in different areas of life than unhappy individuals. For example, happy people tend to have more friends, show fewer signs of mental illness, possess better emotional regulation skills, experience higher life satisfaction, cope with stress more effectively, and experience more positive emotions and fewer negative emotions [23,24]. Therefore, having a higher level of personal happiness can facilitate more effective coping with marital burnout symptoms by promoting the development of mental, psychological, social, and physical resources [23]. Recent research has also shown that happy individuals tend to have higher levels of interpersonal understanding and empathy, spend more quality time with their partners, resolve conflicts more constructively, and have stronger intimacy, sincerity, forgiveness, commitment, and trust in their relationships [16,47]. Thus, happiness may protect married individuals against couple burnout by increasing the frequency of positive emotions (e.g., joy, cheerfulness, and satisfaction) and decreasing the frequency of negative emotions (e.g., anger, cynicism, and anxiety).

Finally, the results showed that the type of marriage, monthly income, education level, and duration of marriage were not significant predictors of couple burnout in married individuals. These findings are consistent with those of some previous studies [2,9,12] but different from others [2,13]. For example, Candemir Karaburç and Tunç [9], Moravejjifar et al. [12], and Nejatian et al. [2] found that the educational level of married individuals was not associated with couple burnout. However, some previous studies have identified an association between couple burnout and factors such as low monthly income [5], duration of marriage [2,12], or type of marriage [13]. It is crucial to acknowledge the significant differences between the cited studies and the current study. For example, Nejatian et al. [2] only focused on married women, and the majority of the independent variables used in their regression model were different from those used in the current study. Moreover, other studies analyzed the independent effects of monthly income, education level, or type of marriage variables and did not examine their effects on the presence of other psychological characteristics of individuals [5,9,13]. Overall, variables such as type of marriage, monthly income, education level, and the duration of marriage may have limited impact on levels of couple burnout, especially when considered with married individuals’ relationship self-efficacy, self-compassion, and happiness levels as well as their gender and the number of children.

### 4.1. Limitations

There are several limitations to consider when interpreting the findings of this study. First, the data for this study were collected from a specific sample of married individuals living in a selected city in the central Black Sea Region of Türkiye. Therefore, the generalizability of the findings to married individuals living in other regions of Türkiye is low. Couple burnout can also occur in different types of relationships; however, only married couples were included in this study. Because couples in different relationship statuses may have different characteristics and dynamics that can affect their experience of couple burnout, the findings cannot be generalized to engaged, dating, or cohabiting couples. Future studies should include couples with different relationship statuses.

A significant limitation of this study is the reliance on self-report scales for data collection. While commonly employed in couple burnout research, self-report measures are susceptible to various biases, including social desirability, mid-point responding, and misconceptions. This is particularly pertinent given the sensitive nature of topics such as relationship self-efficacy and couple burnout. Participants may tend to provide responses they believe are more socially acceptable rather than those that accurately reflect their experiences. Although we utilized reliable measurement tools and allowed the respondents to provide responses anonymously, future studies could benefit from employing multiple methods to overcome these limitations and enhance the validity of the findings. Potential approaches include incorporating physiological measures, conducting interviews, making observations, and using indirect questioning techniques. Additionally, the inclusion of social desirability scales could help identify and mitigate response biases. By diversifying data collection methods, researchers can provide a more comprehensive and accurate representation of couple burnout and related factors in the studied population. Furthermore, a single-item happiness scale was used to measure the happiness levels of the participants. Although the results obtained from previous studies and the current study on the happiness scale provided data that are consistent with the theoretical expectations regarding the convergent and discriminant validity of the scale, only a limited amount of data is available on the reliability of this scale. Therefore, future studies should use multi-item happiness scales to further assess the validity of the findings. Finally, because of the use of a cross-sectional research design, a cause-and-effect relationship cannot be established for the findings. Our study focused primarily on sociodemographic factors, happiness, relationship self-efficacy, and self-compassion. To gain a more comprehensive understanding of couple burnout, future research should consider including control variables such as personality traits (e.g., big-five personality traits), mental health history, caregiving responsibilities, and work–life balance challenges. While our quantitative methodology provided statistically significant findings, it may not fully delineate the complex dynamics underlying couple burnout. To address this limitation, future studies should integrate qualitative methodologies, such as interviews or focus groups, to better elucidate the mechanisms and motivations contributing to couple burnout. This mixed-methods approach can provide a more comprehensive understanding of the phenomenon in the Turkish cultural context.

Another limitation of this study is the generalizability of our findings, particularly in light of potential cultural differences. While our objective was to investigate couple burnout within a Turkish context, we primarily relied on assessment tools originally developed in Western cultures. Although these measures have been validated for use in Turkish samples, it is important to acknowledge that cultural variations, both between countries and within Türkiye, may influence the understanding, perception, and reporting of the constructs under study. The concepts of couple burnout, relationship self-efficacy, and self-compassion may carry subtly different connotations or manifest differently in Turkish culture compared to the Western contexts where these measures were originally developed. Despite the rigorous adaptation and validation processes these instruments have undergone, there remains a possibility that some nuances specific to Turkish marital relationships might not be fully captured. Furthermore, regional differences within Türkiye itself could impact how participants interpret and respond to the scale items. While Samsun is a metropolitan city, it may not fully represent the diversity of marital experiences across all regions of Türkiye. Thus, the findings from this study should be interpreted with caution when generalizing to the broader Turkish population.

### 4.2. Practical Implications

Despite its limitations, the findings of this study provide valuable insights into couple burnout that can be used by mental health professionals and researchers to identify married individuals with couple burnout and develop effective interventions for prevention and treatment. First, our research findings suggest that certain groups such as women, those with a high number of children, and individuals with lower levels of relationship self-efficacy, self-compassion, and happiness may be more vulnerable to couple burnout. Mental health professionals and researchers can use this information to develop screening tools to identify married individuals who are at risk of experiencing high couple burnout. Moreover, training programs to prevent couple burnout could incorporate activities that increase relationship self-efficacy, self-compassion, and happiness levels. These activities can help individuals to develop the skills and resources they need to cope more effectively with the symptoms of couple burnout and the challenges of marriage.

Depending on the needs of married individuals with couple burnout, marriage and family therapists can also carry out activities to increase the relationship self-efficacy, self-compassion, and happiness levels of their clients during individual and group counseling sessions. For example, marriage and family therapists can help clients cope more effectively with couple burnout by providing guidance on developing social and communication skills, asking clients to engage in mindfulness-based practices such as journaling, writing letters to oneself, and cultivating a positive inner voice, and teaching clients positive thinking skills and giving them homework assignments that involve spending time with close friends and engaging in active practices (e.g., walking, physical training, dancing).

## 5. Conclusions

In conclusion, this study examined the relationship between couple burnout and sociodemographic factors, relationship self-efficacy, self-compassion, and happiness in married individuals. The results revealed that being female, having a high number of children, and having low levels of relationship self-efficacy, self-compassion, and happiness were significant positive predictors of couple burnout among Turkish married individuals. The type of marriage, monthly income, and duration of marriage were not predictors of couple burnout. Given the limited research on couple burnout in diverse cultural contexts, this study may help identify universal and culture-specific factors related to couple burnout among married individuals. The research findings may also help researchers achieve a more comprehensive understanding of the role of sociodemographic factors, relationship self-efficacy, self-compassion, and happiness in couple burnout.

## Figures and Tables

**Table 1 behavsci-14-00561-t001:** Results of the Pearson product-moment correlation analyses.

					Couple Burnout
Variables	Minimum	Maximum	*M*	*SD*	*r*
1. Couple burnout	1.00	6.60	2.80	1.33	
2. Gender	0.00	1.00	0.65	0.48	0.22 **
3. Type of marriage	0.00	1.00	0.14	0.35	0.08
4. Monthly income	0.00	1.00	0.68	0.47	−0.18 **
5. High-school or below education	0.00	1.00	0.19	0.39	0.09
6. Associate degree	0.00	1.00	0.12	0.33	−0.00
7. Undergraduate degree	0.00	1.00	0.54	0.50	−0.05
8. Graduate degree	0.00	1.00	0.15	0.36	−0.02
9. 1–5 years of marriage	0.00	1.00	0.22	0.42	−0.10
10. 6–10 years of marriage	0.00	1.00	0.29	0.46	−0.07
11. 11–15 years of marriage	0.00	1.00	0.20	0.40	0.13 *
12. 16–20 years of marriage	0.00	1.00	0.12	0.33	0.16 *
13. 21 years or more of marriage	0.00	1.00	0.16	0.37	−0.09
14. Number of children	0.00	3.00	1.49	0.86	0.22 **
15. Relationship self-efficacy	32.00	80.00	64.45	8.90	−0.42 **
16. Self-compassion	49.00	115.00	80.86	12.29	−0.38 **
17. Happiness	1.00	10.00	7.44	1.70	−0.66 **

Note. Variables were coded as follows: Gender: 0 = men, 1 = women. Type of marriage: 0 = choice, 1 = parent-arranged. Monthly income: 0 = TRY 7000 and below, 1 = TRY 7001 and above. High-school or below education level: 0 = other education levels, 1 = high-school or below. Associate degree: 0 = other education levels, 1 = associate degree. Undergraduate degree: 0 = other education levels, 1 = undergraduate degree. Graduate degree: 0 = other education levels, 1 = graduate degree. Duration of marriage: 0 = other marriage durations, 1 = 1–5 years of marriage; 6–10 years of marriage: 0 = other marriage durations, 1 = 6–10 years of marriage; 11–15 years of marriage: 0 = other marriage durations, 1 = 11–15 years of marriage; 16–20 years of marriage: 0 = other marriage durations, 1 = 16–20 years of marriage; 21 years or more of marriage: 0 = other marriage durations, 1 = 21 years or more of marriage. *p* ˂ 0.01 *, *p* ˂ 0.001 **.

**Table 2 behavsci-14-00561-t002:** The change statistics for standard multiple linear regression analysis.

Model	*R*	*R* ^2^	Adj. *R*^2^	*SE Est.*	Change Statistics				
					Δ*R*^2^	Δ*F*	*df* _1_	*df_2_*	*p*
Couple burnout	0.74	0.55	0.53	0.91	0.55	33.62	14	386	0.001 **

Note. *p* ˂ 0.001 **.

**Table 3 behavsci-14-00561-t003:** Results of standard multiple linear regression and relative importance analyses of couple burnout scores.

	*B*	*SE*	β	*t*	*p*	*sr*	*d_p_*
Intercept	8.09	0.46		17.69	0.001 **		
Gender	0.44	0.10	0.16	4.44	0.001 **	0.15	0.06
Type of marriage	−0.11	0.14	−0.03	−0.78	0.437	−0.03	−0.00
Monthly income	−0.20	0.11	−0.07	−1.83	0.068	−0.06	0.02
Associate	0.11	0.17	0.03	0.66	0.513	0.02	−0.00
Undergraduate	0.01	0.13	0.00	0.07	0.944	0.00	−0.00
Graduate	0.01	0.17	0.00	0.07	0.946	0.00	−0.00
6–10 years of marriage	−0.23	0.15	−0.08	−1.54	0.125	−0.05	0.00
11–15 years of marriage	−0.08	0.17	−0.02	−0.46	0.649	−0.02	−0.01
16–20 years of marriage	0.18	0.20	0.04	0.92	0.358	0.03	0.01
21 years or more marriage	−0.25	0.19	−0.07	−1.31	0.191	−0.05	0.01
Number of children	0.22	0.07	0.14	3.14	0.002 *	0.11	0.06
Relationship self-efficacy	−0.02	0.01	−0.16	−4.01	0.001 **	−0.14	0.12
Self-compassion	−0.01	0.00	−0.13	−3.32	0.001 **	−0.11	0.09
Happiness	−0.41	0.03	−0.52	−12.97	0.001 **	−0.44	0.62

Note. Variables were coded as follows: Gender: 0 = men, 1 = women. Type of marriage: 0 = choice, 1 = parent-arranged. Monthly income: 0 = TRY 7000 and below, 1 = TRY 7001 and above. For the education level variable, the high-school and below education level was used as the reference category. For the marriage duration variable, 1–5 years of marriage duration was used as the reference category. *sr* = semi-partial correlation. *dp* = Pratt’s relative importance index value. *p* ˂ 0.01 *, *p* ˂ 0.001 **.

## Data Availability

Data underlying this article are available from the Open Science Framework (osf.io/sb7xp).

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
