# Peer review of "Predictors of Couple Burnout among Turkish Married Individuals"

_behavsci, 2024, doi:10.3390/bs14070561_

Round 1

Reviewer 1 Report

Comments and Suggestions for Authors

The study provides an overview of the importance of investigating couple burnout, emphasizing the lack of research conducted in the Turkish setting. The background is well-organized, it includes pertinent studies and discerns the necessity for the present investigation.

The method provides a comprehensive description of the sample demographics. Nevertheless, additional details regarding the recruitment procedure would be beneficial. However, adding additional control variables like personality traits, mental health history, and external stressors may have improved it. The study's quantitative approach lacks some of the insight that qualitative research could provide. Integrating qualitative methodologies such as interviews or focus groups would enhance comprehension of the mechanisms and motivations behind the contribution of certain elements to couple burnout within the Turkish setting. Furthermore, why do authors consider that Samsun, is representative for the purpose of the study, may be a useful comment.

One limitation of the study is the issue of generalizability, as cultural differences can still impact how concepts are understood and communicated. Although the study's objective is to investigate relationship burnout within a Turkish context, it mainly depends on assessment tools that were originally established in Western cultures. While the validity of these measures has been confirmed in Turkish samples, it is important to consider that cultural variations, including those within the country, may potentially impact the perception and reporting of constructs. The study should approach or limitations must be in place to evidentiate potential cultural biases in the interpretation of findings.

Although the study employs reliable measurement tools, it is important to recognize the possibility of response bias, particularly in self-reported assessments of sensitive subjects such as relationship self-efficacy and couple burnout. This bias may arise due to individuals' tendency to provide socially desirable responses. I would like to emphasize that the inclusion of techniques to reduce or identify such biases would have enhanced the study.

The predictors analyzed encompass sociodemographic characteristics, relational self-efficacy, happiness, and self-compassion. I appreciate that multiple regression is an appropriate statistical method for analyzing the impact of multiple predictor factors on a single result variable. This approach enables the study to concurrently manage multiple parameters and evaluate the distinct impact of each predictor on couple burnout. The data supports the conclusions and it emphasizes the significance of addressing couple burnout with specific interventions.

In my opinion, this study provides a contribution to understanding couple burnout in Türkiye culture and provides a good basis for future research and intervention development and these minor suggestions could only bring further clarifications.

Author Response

Comment#1. The study provides an overview of the importance of investigating couple burnout, emphasizing the lack of research conducted in the Turkish setting. The background is well-organized, it includes pertinent studies and discerns the necessity for the present investigation.

Thanks for this positive comment. There is nothing to change for this comment. 

Comment#2. The method provides a comprehensive description of the sample demographics. Nevertheless, additional details regarding the recruitment procedure would be beneficial.

Thanks for this helpful comment. We added following to Setting section for more information about recruitment process. Yellow highlighted in text.

Participants were recruited using a combination of online and in-person methods. Online announcements were posted on popular social media platforms and forums (e.g., Instagram, X, and Facebook) targeting married individuals. We recruited individuals for face-to-face data from local community centers, universities, other public areas, and word-of-mouth referrals. Additionally, recruitment messages were sent to local organizations and community groups. Eligible participants were required to be currently married, at least 17 years old, fluent in Turkish, and residing in Samsun. 

Comment#3. However, adding additional control variables like personality traits, mental health history, and external stressors may have improved it. The study's quantitative approach lacks some of the insight that qualitative research could provide. Integrating qualitative methodologies such as interviews or focus groups would enhance comprehension of the mechanisms and motivations behind the contribution of certain elements to couple burnout within the Turkish setting.

We added following statements to limitations as per reviewer suggestion. We added following to Limitations Section.

Our study focused primarily on sociodemographic factors, happiness, relationship self-efficacy, and self-compassion. To gain a more comprehensive understanding of couple burnout, future research should consider including control variables such as personality traits (e.g., big-five personality traits), mental health history, caregiving responsibilities and work-life balance challenges. While our quantitative methodology provided statistically significant findings, it may not fully delineate the complex dynamics underlying couple burnout. To address this limitation, future studies should integrate qualitative methodologies, such as interviews or focus groups, to better elucidate the mechanisms and motivations. This mixed-methods approach can provide a more comprehensive understanding of the phenomenon in the Turkish cultural context.

Comment#4. Furthermore, why do authors consider that Samsun, is representative for the purpose of the study, may be a useful comment.

We added following statements to Setting Section as per reviewer suggestion. We added following to Setting Section.

As a metropolitan city, Samsun has a diverse population in terms of socioeconomic status, education levels, and occupations, which allows for a varied sample of married individuals.

Comment #5. One limitation of the study is the issue of generalizability, as cultural differences can still impact how concepts are understood and communicated. Although the study's objective is to investigate relationship burnout within a Turkish context, it mainly depends on assessment tools that were originally established in Western cultures. While the validity of these measures has been confirmed in Turkish samples, it is important to consider that cultural variations, including those within the country, may potentially impact the perception and reporting of constructs. The study should approach or limitations must be in place to evidentiate potential cultural biases in the interpretation of findings.

We added following statements to limitations as per reviewer suggestion. We added following to Limitations Section.

Another limitation of this study is the generalizability of our findings, particularly in light of potential cultural differences. While our objective was to investigate couple burnout within a Turkish context, we primarily relied on assessment tools originally developed in Western cultures. Although these measures have been validated for use in Turkish samples, it is important to acknowledge that cultural variations, both be-tween countries and within Türkiye, may influence the understanding, perception, and reporting of the constructs under study. The concepts of couple burnout, relationship self-efficacy, and self-compassion may carry subtly different connotations or manifest differently in Turkish culture compared to the Western contexts where these measures were originally developed. Despite the rigorous adaptation and validation processes these instruments have undergone, there remains a possibility that some nuances specific to Turkish marital relationships might not be fully captured. Furthermore, regional differences within Türkiye itself could impact how participants interpret and respond to the scale items. While Samsun is a metropolitan city, it may not fully represent the diversity of marital experiences across all regions of Türkiye. Thus, the findings from this study should be interpreted with caution when generalizing to the broader Turkish population.

Comment#6. Although the study employs reliable measurement tools, it is important to recognize the possibility of response bias, particularly in self-reported assessments of sensitive subjects such as relationship self-efficacy and couple burnout. This bias may arise due to individuals' tendency to provide socially desirable responses. I would like to emphasize that the inclusion of techniques to reduce or identify such biases would have enhanced the study.

We added following statements to limitations as per reviewer suggestion. We added following to Limitations Section.

A significant limitation of this study is the reliance on self-report scales for data collection. While commonly employed in couple burnout research, self-report measures are susceptible to various biases, including social desirability, mid-point responding, and misconceptions. This is particularly pertinent given the sensitive nature of topics such as relationship self-efficacy and couple burnout. Participants may tend to provide responses they believe are more socially acceptable rather than those that accurately reflect their experiences. Although we utilized reliable measurement tools and allowed to provide responses anonymously to participants, future studies could benefit from employing multiple methods to overcome these limitations and enhance the validity of findings. Potential approaches include incorporating physiological measures, conducting interviews, making observations, and using indirect questioning techniques. Additionally, the inclusion of social desirability scales could help identify and mitigate response biases. By diversifying data collection methods, researchers can provide a more comprehensive and accurate representation of couple burnout and related factors in the studied population.

Comment#7. The predictors analyzed encompass sociodemographic characteristics, relational self-efficacy, happiness, and self-compassion. I appreciate that multiple regression is an appropriate statistical method for analyzing the impact of multiple predictor factors on a single result variable. This approach enables the study to concurrently manage multiple parameters and evaluate the distinct impact of each predictor on couple burnout. The data supports the conclusions, and it emphasizes the significance of addressing couple burnout with specific interventions.

Thanks for this positive comment. There is nothing to change for this comment.

Comment#8. In my opinion, this study provides a contribution to understanding couple burnout in Türkiye culture and provides a good basis for future research and intervention development and these minor suggestions could only bring further clarifications.

Thanks for this positive comment. There is nothing to change for this comment.

Comment#9. English language fine. No issues detected.

Thanks for this positive comment. There is nothing to change for this comment.

Reviewer 2 Report

Comments and Suggestions for Authors

This is a very good paper, overall. However, there are areas where the quality can be much more enhanced. For example, it might be very useful to indicate the value the findings add to what we already know about this subject, This should be added in the last line of the abstract.

Comments on the Quality of English Language

The language is good enough based on the fact the authors are not speakers of the English Language. For example, there were instances where the words "although" and "however" have been used in the same sentence. Such usages usually render the meaning being conveyed difficult. See one instance in Line 92.

Author Response

Comment#1. This is a very good paper, overall. However, there are areas where the quality can be much more enhanced. For example, it might be very useful to indicate the value the findings add to what we already know about this subject, This should be added in the last line of the abstract.

Thanks for this helpful comment. We added following to Abstract section Line 219-226 for more information about recruitment process. Yellow highlighted in text.

The research findings demonstrate the multidimensional nature of couple burnout and provide a more nuanced understanding of its predictive factors. These results have potential implications for the development of evidence-based and targeted interventions in relationship education programs.

Comment#2. The language is good enough based on the fact the authors are not speakers of the English Language. For example, there were instances where the words "although" and "however" have been used in the same sentence. Such usages usually render the meaning being conveyed difficult. See one instance in Line 92.

A proficient English speaker read to manuscript and corrected. Moreover, we changed Line 92 sentence and removed However, we also checked along the text and reduced sentences with However.  Yellow highlighted in text. New form looks as follows:

Although there is a growing body of research on the relationship between general self-efficacy and burnout, there is a paucity of research on the relationship between relationship self-efficacy and couple burnout [12].
